# REGULARIZING ENERGY AMONG TRAINING SAMPLES FOR OUT-OF-DISTRIBUTION GENERALIZATION

**Yiting Chen**[1]**, Qitian Wu**[2]*& **Junchi Yan**[1] [†]
[1]Sch. of Computer Science & Sch. of Artificial Intelligence, Shanghai Jiao Tong University
[2]Eric and Wendy Schmidt Center, Broad Institute of MIT and Harvard
{sjtucyt, yanjunchi}@sjtu.edu.cn, wuqitian@broadinstitute.org

## ABSTRACT

The energy-based model provides a unified framework for various learning models where an energy value is assigned to each configuration of random variables based on probability. Recently, different methods have been proposed to derive an energy value out of the logits of a classifier for out-of-distribution (OOD) detection or OOD generalization. However, these methods mainly focus on the energy difference between in-distribution and OOD data samples, neglecting the energy difference among in-distribution data samples. In this paper, we show that the energy among in-distribution data also requires attention. We propose to investigate the energy difference between in-distribution data samples. Both empirically and theoretically, we show that previous methods for subpopulation shift (*e.g.*, long-tail classification) such as data re-weighting and margin control apply implicit energy regularization and we provide a unified framework from the energy perspective. With the influence function, we further extend the energy regularization framework to OOD generalization scenarios where the distribution shift is more implicit compared to the long-tail recognition scenario. We conduct experiments on long-tail datasets, subpopulation shift benchmarks, and OOD generalization benchmarks to show the effectiveness of the proposed energy regularization.

## 1 INTRODUCTION

Energy-based models (EBMs) (LeCun et al., 2006; Ranzato et al., 2006; 2007) provide a unified theoretical framework for various learning models where an energy value is assigned to each configuration of random variables regarding its probability. Derived from the logits of a classifier, previous works show that a discriminative modpel is also an energy model (Xie et al., 2016; Grathwohl et al., 2020) and use it for generative tasks. Inspired by this, the corresponding energy model is employed for OOD detection (Liu et al., 2020; Bitterwolf et al., 2022; Wu et al., 2023). Out-of-distribution (OOD) data samples are detected by a higher energy value than in-distribution(IID) data samples due to the classifier's low probability of seeing out-of-distribution data samples. Recent work (Xie et al., 2022) also minimizes the distance of energy distribution between the source domain and target domain to enhance the performance in domain adaptation. However, these previous works mainly focus on the energy difference between IID data and OOD data, neglecting the energy difference between in-distribution data samples.

In this paper, we propose to investigate the energy difference between in-distribution data samples (training data samples) and regularize the energy on training samples to boost the OOD generalization performance. From the energy regularization perspective, we both empirically and theoretically show that long-tail recognition methods (Wang et al., 2017; Zhou et al., 2018; Liu et al., 2019; Zhong et al., 2019; He et al., 2021; Zhong et al., 2021) (a special case of sub-population shift (Cai et al., 2021; Koh et al., 2021)) such as reweighting data (Zhang et al., 2018; Zhao et al., 2019; Ye et al., 2020; Hsieh et al., 2021) or controlling the classification margin (Cao et al., 2019) could be

---

*The work was mostly completed when Qitian Wu was a Ph.D. student with SJTU

†Correspondence author who is also affiliated with Shanghai Artificial Intelligence Laboratory. The SJTU authors were partly supported by NSFC (62222607), Shanghai Municipal Science and Technology Major Project (2021SHZDZX0102), and Seed fund of SJTU-UCL joint research program.

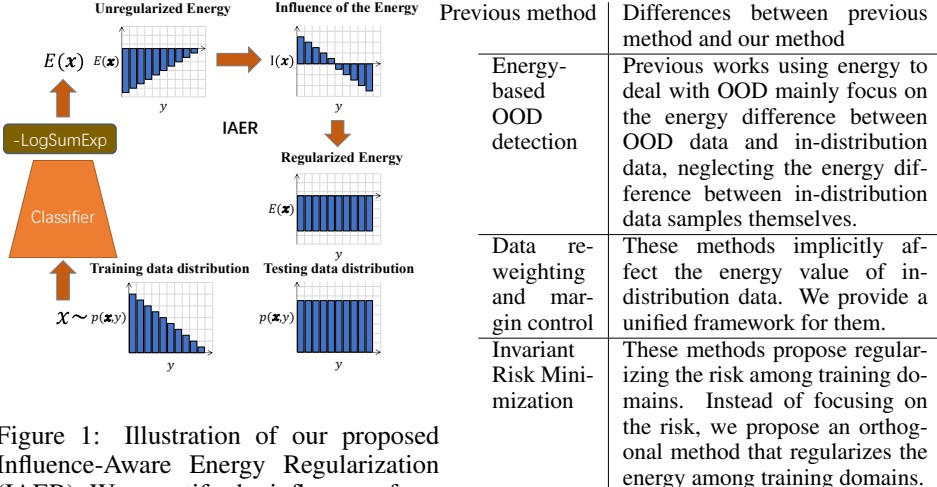

Figure 1: Illustration of our proposed Influence-Aware Energy Regularization (IAER). We quantify the influence of energy regularization and apply regularization accordingly.

| Previous method | Differences between previous method and our method |
|---|---|
| Energy-based OOD detection | Previous works using energy to deal with OOD mainly focus on the energy difference between OOD data and in-distribution data, neglecting the energy difference between in-distribution data samples themselves. |
| Data re-weighting and margin control | These methods implicitly affect the energy value of in-distribution data. We provide a unified framework for them. |
| Invariant Risk Minimization | These methods propose regularizing the risk among training domains. Instead of focusing on the risk, we propose an orthogonal method that regularizes the energy among training domains. |

Table 1: Some previous works and ours

regarded as an implicit regularization on the energy value of training samples. For domain generalization, a main branch of works focuses on invariant risk minimization (Chang et al., 2020; Creager et al., 2021; Lin et al., 2021), which regularizes the risk across different training domains to learn an invariant classifier. While the energy value is unrelated to the risk, we show that regularizing energy value across training domains could also improve the OOD generalization and is orthogonal to the methods for invariant risk minimization. We propose an energy regularization method and verify its effectiveness on tasks in long-tail classification, subpopulation shift, and domain generalization. Other empirical findings regarding the energy distribution of training data samples are provided. **We summarize the contributions of this paper in the following:**

- Besides the extensive research for OOD detection and OOD generalization focusing on the energy difference between out-of-distribution data and in-distribution data (Liu et al., 2020; Bitterwolf et al., 2022; Wu et al., 2023; Xie et al., 2022), to the best of our knowledge, we are the first to call for attention to the energy difference between in-distribution data samples.
- We theoretically show that data re-weighting and margin control, the two branches of methods proposed for long-tail classification (a case for subpopulation shift) could be unified as implicit energy regularization among in-distribution data samples.
- We propose a method namely influence aware energy regularization to regularize energy among training domains to boost the model performance on OOD generalization.

## 2 RELATED WORKS

**Energy Based Learning.** Energy-based models (EBMs) (LeCun et al., 2006; Ranzato et al., 2006; 2007) provide a unified theoretical framework for various learning models. Recent works employ the energy function defined on discriminative models for other tasks *e.g.* generative learning or OOD detection. Xie et al. (2016) shows that a generative random field model can be derived from a discriminative neural network. While Grathwohl et al. (2020) finds that neural classifiers are also energy-based models for joint distribution and devises a hybrid model that acts as both discriminative and generative models. Liu et al. (2020) proposes to use the energy value to detect OOD samples, which has been theoretically proven (Bitterwolf et al., 2022) to be equal to training an additional binary discriminator. Recent work (Xie et al., 2022) minimizes the distance of energy distribution between the source domain and target domain to enhance the performance in domain adaptation (see Wang & Deng (2018) and the references therein). However, these previous works focus on the energy difference between in-distribution data and out-of-distribution data, and we show that the energy difference between in-distribution data points could influence the generalization.

**Long-Tail Recognition.** Long-tail recognition has drawn increasing attention (Wang et al., 2017; Zhou et al., 2018; Liu et al., 2019; Zhong et al., 2019; He et al., 2021; Zhong et al., 2021) due

to the pervasiveness of imbalanced data in real-world scenarios. Most methods could be divided into three categories: re-sampling the data, re-weighting the loss, and transfer learning. For re-sampling, various methods have been proposed to re-sample the dataset for a more balanced data distribution (Chawla et al., 2002; Estabrooks et al., 2004; Han et al., 2005; Liu et al., 2009; Shen et al., 2016; Liu et al., 2019; Wang et al., 2020; Kang et al., 2020; Zhang & Pfister, 2021). As for re-weighting, re-weighting methods assign different losses to different classes (Zhang et al., 2018; Zhao et al., 2019; Ye et al., 2020; Hsieh et al., 2021) or different data samples (Lin et al., 2017; Ren et al., 2018; Shu et al., 2019) to balance the performance on each class. Specifically, LDAM (Cao et al., 2019) proposes a distribution-aware loss that enlarges the margin to less frequent (tail) classes.

**Subpopulation Shift.** Subpopulation Shift focuses on changing the proportion of some subpopulations (Cai et al., 2021; Koh et al., 2021), where subpopulations refer to subsets of a data domain divided by certain attributes. A conventional setting defines subpopulations as the product of attributes and classes (Geirhos et al., 2020). In fact, long-tail recognition is a special case of subpopulation shift. Similar to long-tail classification, models tend to learn spurious features when minimizing overall loss, resulting in poor performance on minority subpopulations (DeGrave et al., 2021; Joshi et al., 2022). A wide array of methods has been developed, some focusing on scenarios where attributes are known (Gowda et al., 2021; Izmailov et al., 2022; Menon et al., 2020; Nam et al., 2022; Sagawa et al., 2019; Yao et al., 2022), while others investigate cases where attributes are unknown (Creager et al., 2021; Han et al., 2022; Idrissi et al., 2022; Liu et al., 2021a).

**Domain Generalization.** Specifically, domain generalization (DG) (Blanchard et al., 2011) aims to train a model using data from a single or multiple source domain that would generalize well to any out-of-distribution(OOD) target domains. Various methods have been proposed to tackle the domain generalization problem including learning domain-invariant representations (Muandet et al., 2013; Li et al., 2018b;c), augmenting the data (Zhou et al., 2020; Yan et al., 2020) and applying meta-learning to domain generalization (Li et al., 2018a; Balaji et al., 2018). Some of the methods propose regularization strategies designed based on heuristics that surpass the predictive power of an auxiliary CNN implemented as a stack of $1 \times 1$ convolution layers (Wang et al., 2019) or mask out the features with large gradients (Huang et al., 2020). See Wang et al. (2021); Zhou et al. (2021) for a comprehensive survey.

## 3 ASSOCIATING METHODS IN LONG-TAIL RECOGNITION WITH TRAINING ENERGY REGULARIZATION

Various methods for long-tail recognition (Wang et al., 2017; Zhou et al., 2018; Liu et al., 2019; Zhong et al., 2019; He et al., 2021; Zhong et al., 2021) (a case of subpopulation shift (Cai et al., 2021; Koh et al., 2021)) have been proposed with different motivations. In this section, we show that these different long-tail recognition methods implicitly change the training energy and that energy regularization among training data samples unifies two different branches of long-tail recognition methods (reweighting (Zhang et al., 2018; Zhao et al., 2019; Ye et al., 2020; Hsieh et al., 2021) and margin control (Cao et al., 2019)).

### 3.1 METHODS FOR LONG-TAIL RECOGNITION ARE IMPLICIT ENERGY REGULARIZATION

For a $K$-class classification problem, a parameterized classifier $f_\theta : \mathbb{R}^D \to \mathbb{R}^K$ maps data point $\mathbf{x} \in \mathbb{R}^D$ to $K$ real-valued logits where $\theta$ is the trainable parameter. For a data point $\mathbf{x}$ and its corresponding label $y$, the loss for the parameter $\theta$ is defined as $\mathcal{L}(\mathbf{x}, y, \theta)$. Energy-based model (LeCun et al., 2006; Grathwohl et al., 2020) $E(\mathbf{x}) : \mathbb{R}^D \to \mathbb{R}$ maps each data point $\mathbf{x}$ to a single, non-probabilistic scalar called the energy, where the energy value could be turned to a probability as:

$$p(\mathbf{x}) = \frac{e^{-E(\mathbf{x})/T}}{\int_{\mathbf{x}'} e^{-E(\mathbf{x}')/T}}. \tag{1}$$

For classifier $f_\theta$, the logits are typically converted to a normalized probability distribution with the Softmax function: $\bar{p}_\theta(y|\mathbf{x}) = \frac{\exp(f_\theta(\mathbf{x})[y])}{\sum_{y'} \exp(f_\theta(\mathbf{x})[y'])}$, where $f(\mathbf{x})[y]$ represents the logit corresponding to the $y$-th class. The joint distribution of data $\mathbf{x}$ and label $y$ could be defined as $\bar{p}_\theta(\mathbf{x}, y) = \frac{\exp(f_\theta(\mathbf{x})[y])}{Z(\theta)}$ where $Z(\theta)$ is unknown normalizing constant. By marginalizing out $y$, the unnormalized density

model for $\mathbf{x}$ is $\bar{p}_\theta(\mathbf{x}) = \sum_{y'} \bar{p}_\theta(\mathbf{x}, y') = \frac{\sum_{y'} \exp(f_\theta(\mathbf{x})[y'])}{Z(\theta)}$. Therefore the energy at data point $\mathbf{x}$ regarding to $\bar{p}_\theta(\mathbf{x})$ could be defined as:

$$E_\theta(\mathbf{x}) = -\log \sum_{y'} \exp\left(f_\theta(\mathbf{x})[y']\right). \tag{2}$$

The energy defined on classifiers is firstly introduced for generative tasks (Xie et al., 2016; Grathwohl et al., 2020), where a classifier could be treated as an EBM and used for image generation. The energy of classifiers is also used for OOD detection (Liu et al., 2020; Bitterwolf et al., 2022; Wu et al., 2023), where unseen data samples (OOD samples) generally have higher energy.

Similar to the OOD detection scenario, the energy is lower on sub-populations with large amounts of data in the sub-population shift scenarios (Cai et al., 2021; Koh et al., 2021) (*e.g.* the classes having much more data samples than other classes in long-tail classification). In this section, taking long-tail recognition as a typical and clear case of sub-population shift, we show that various methods proposed to tackle the long-tail recognition problem apply implicit energy regularization.

Data resampling or re-weighting (Zhang et al., 2018; Zhao et al., 2019; Ye et al., 2020; Hsieh et al., 2021) is one of the main branches of methods for long-tail recognition. By assigning different weights to different classes or sampling the data from the minor classes with little data more often, these methods encourage the model to reduce the loss of training data samples from the minor classes. Similar to the OOD detection scenario, the energy on more frequently trained data would be lower, which implicitly regularizes the energy on training data samples. Cao et al. (2019) also proposes margin control for better long-tail recognition performance. The classification margin is the difference between the logit of the ground-truth label and the largest logit of non-ground-truth labels. By controlling the classification margin to be larger for data samples from minor classes, the method prevents the model from misclassifying minor classes and improves the model performance. By controlling the margin, the logits are enlarged for the data samples from minor classes, which also implicitly regularizes the energy and encourages a more uniform energy distribution between the minor and major classes.

In Fig. 2, we train ResNet-32 on CIFAR10-LT and CIFAR100-LT following Cao et al. (2019) and report the average energy of each class. Generally, for the models trained with ERM (Vapnik, 1998), the energy is lower on the classes with more data samples, indicating a larger probability $p(\mathbf{x})$ (Pearson Correlation Coefficient is at $-0.74$ on CIFAR10-LT and $-0.60$ on CIFAR100-LT). This phenomenon is because the model is trained to give lower energy (corresponding to higher $\bar{p}(\mathbf{x})$) to the frequently trained data, which has been utilized to detect OOD samples (Liu et al., 2020). On the other hand, the models trained with LDAM have a more uniform energy distribution among classes (Pearson Correlation Coefficient is at $-0.26$ on CIFAR10-LT and $0.16$ on CIFAR100-LT. It empirically shows how the margin control implicitly affects the energy among training samples. In the next section, we theoretically unify the sample re-weighting and margin control from the energy regularization perspective.

As a special case of sub-population shift, the long-tail recognition shows a clear probability shift as the probability of data samples from some classes is higher. Methods such as re-weighting or margin control implicitly affect the energy, pushing the $p(\mathbf{x})$ predicted by the model closer to following a uniform distribution, the ground truth distribution in the test set.

### 3.2 Unifying Reweighting and Margin Control in Energy Regularization

To deal with the imbalanced distribution among different classes in long-tail recognition, many previous works resort to assigning different weights to samples from different classes (Zhang et al., 2018; Zhao et al., 2019; Ye et al., 2020; Hsieh et al., 2021), while other works such as LDAM (Cao et al., 2019) adjust the margin to the decision boundary for different classes. In this section, we show that energy regularization actually unifies both data re-weighting and margin control. Starting with a cross-entropy loss, for a data sample $\mathbf{x}$ and the ground truth label $y$, we have

$$\mathcal{L}_{ce}(\mathbf{x}, y, \theta) = -\log \frac{\exp(f_\theta(\mathbf{x})[y])}{\sum_{y'} \exp(f_\theta(\mathbf{x})[y'])} = -f_\theta(\mathbf{x})[y] - E_\theta(\mathbf{x}) \tag{3}$$

Adding an energy regularization with coefficient $\hat{\beta}_\mathbf{x}$ regarding the input $\mathbf{x}$, we have

$$\mathcal{L}_{ce}(\mathbf{x}, y, \theta) + \hat{\beta}_\mathbf{x} \cdot E_\theta(\mathbf{x}) = -f_\theta(\mathbf{x})[y] - (1 - \hat{\beta}_\mathbf{x}) \cdot E_\theta(\mathbf{x}) \tag{4}$$

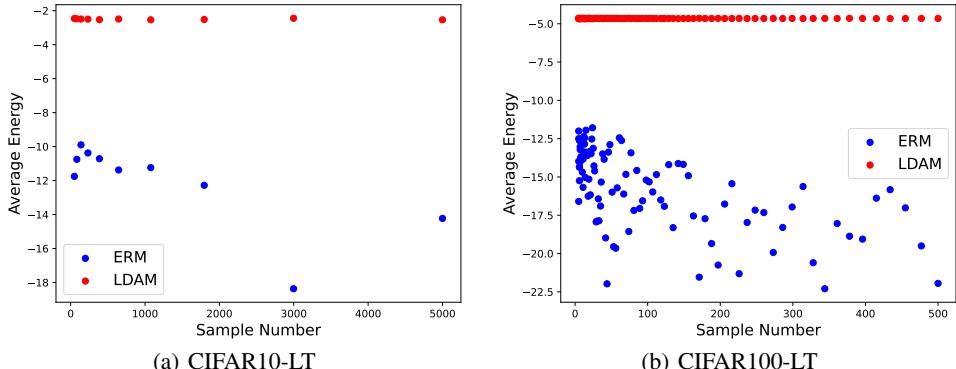

(a) CIFAR10-LT                    (b) CIFAR100-LT

Figure 2: Average energy of each class of ResNet-32 trained on CIFAR10-LT and CIFAR100-LT. We follow the hyper-parameter setting in Cao et al. (2019) to train models. Each dot corresponds to a class. The model trained by ERM has a biased energy distribution, where energy is generally lower for the class with more data. (The Pearson correlation coefficient is $-0.74$ on CIFAR10-LT and $-0.60$ on CIFAR100-LT.) As for the model trained with LDAM (Cao et al., 2019), the average energy of different classes is nearly identical. (The Pearson correlation coefficient is $-0.26$ on CIFAR10-LT and $0.16$ on CIFAR100-LT.) The results empirically indicate the LDAM is actually an implicit energy regularization and levels the energy on samples from different classes.

Then the gradient of the loss is:

$$\frac{\partial[\mathcal{L}_{ce}(\mathbf{x}, y, \theta) + \hat{\beta}_{\mathbf{x}} \cdot E_\theta(\mathbf{x})]}{\partial \theta} = \left[(1 - \hat{\beta}_{\mathbf{x}}) \cdot \bar{p}(y|\mathbf{x}) - 1\right] \frac{\partial f_\theta(\mathbf{x})[y]}{\partial \theta} + (1 - \hat{\beta}_{\mathbf{x}}) \sum_{y' \neq y} \bar{p}(y'|\mathbf{x}) \cdot \frac{\partial f_\theta(\mathbf{x})[y']}{\partial \theta}.$$

(5)

When $\hat{\beta}_{\mathbf{x}} \neq 1$, we could further derive the gradient as:

$$\frac{\partial \mathcal{L}(\mathbf{x}, y, \theta)}{\partial \theta} = (1 - \hat{\beta}_{\mathbf{x}}) \cdot \left[\bar{p}(y|\mathbf{x}) - \frac{1}{1 - \hat{\beta}_{\mathbf{x}}}\right] \frac{\partial f_\theta(\mathbf{x})[y]}{\partial \theta} + (1 - \hat{\beta}_{\mathbf{x}}) \cdot \sum_{y' \neq y} \bar{p}(y'|\mathbf{x}) \cdot \frac{\partial f_\theta(\mathbf{x})[y']}{\partial \theta}.$$ (6)

where $\mathcal{L}(\mathbf{x}, y, \theta) = \mathcal{L}_{ce}(\mathbf{x}, y, \theta) + \hat{\beta}_{\mathbf{x}} \cdot E_\theta(\mathbf{x})$. As shown in Eq. 6, the influence of energy regularization is twofold: adjusting the margin and reweighting data points. The weight for each data point is $1 - \hat{\beta}_{\mathbf{x}}$. As for the margin, the margin is defined as $f_\theta(x)[y] - max_{j \neq y} f_\theta(x)[j]$ in Cao et al. (2019). In Eq. 6, the coefficient of the gradient $\frac{\partial f_\theta(\mathbf{x})[y]}{\partial \theta}$ is changed from $\bar{p}(y|\mathbf{x}) - 1$ to $\bar{p}(y|\mathbf{x}) - \frac{1}{1 - \hat{\beta}_{\mathbf{x}}}$. When $0 < \hat{\beta}_{\mathbf{x}} < 1$, the energy regularization enlarges the margin by pushing down the coefficient of $\frac{\partial f_\theta(\mathbf{x})[y]}{\partial \theta}$ and down-weights the data point $\mathbf{x}$ with coefficient $(1 - \hat{\beta}_{\mathbf{x}})$. When $\hat{\beta}_{\mathbf{x}} < 0$, the regularizer reduces the margin and up-weights the data $\mathbf{x}$ with coefficient $(1 - \hat{\beta}_{\mathbf{x}})$. Therefore, we theoretically show that regularizing energy is actually a combination of data reweighting and margin control unifying two different branches of methods in long-tail recognition.

## 4    REGULARIZING ENERGY FOR OOD GENERALIZATION

In Sec. 3, we show that long-tail recognition methods such as LDAM (Cao et al., 2019) implicitly regularize the energy. In this section, we want to extend the energy regularization to a more general OOD generalization scenario. In the long-tail recognition scenario, the disparity of data amount between different classes is known, guiding the data re-weighting or margin control. Since the distribution shift in the OOD generalization scenario is more implicit, it requires a method to determine the energy regularization coefficient $\hat{\beta}_{\mathbf{x}}$. Note that energy regularization is orthogonal to the previous methods regularizing the risk among different domains the reasons are as follows.

**Motivation:** The energy correlates to the probability of a data sample $p(\mathbf{x})$ predicted by the model. Similar to the long-tail recognition scenario, we wish the predicted probability to be close to the

actual probability in the test domain. While we have no way of knowing the test data distribution, the least we can do is to prevent the model from being over-confident (*i.e.* expect the unexpected). Note that the energy does not correspond to the prediction of the classifier (Remark 4.1), which is the reason most previous works overlook the energy disparity between training data samples and also make the energy regularization orthogonal to previous works focusing on regularizing the risk (Chang et al., 2020; Creager et al., 2021; Lin et al., 2021).

**Remark 4.1.** *[Arbitrary Energy] Consider $\forall (\mathbf{x}, y) \in D_{train}$ and a classifier $f_\theta$. For $\forall \mathcal{E} \in \mathbb{R}$, there exists a classifier $g_\eta$ that satisfy*

$$
\begin{aligned}
\bar{p}_\theta(y|\mathbf{x}) &= \bar{p}_\eta(y|\mathbf{x}), \\
E_\eta(\mathbf{x}) &= \mathcal{E}.
\end{aligned}
\tag{7}
$$

*where $\bar{p}_\theta(y|\mathbf{x})$ and $\bar{p}_\eta(y|\mathbf{x})$ is the conditional probability predicted by $f_\theta$ and $g_\eta$ respectively while $E_\eta(\mathbf{x})$ is the energy value of $g_\eta$ on data point $\mathbf{x}$.*

## 4.1 DETERMINING ENERGY REGULARIZATION COEFFICIENT VIA INFLUENCE FUNCTION

The difference between a typical OOD generalization scenario and long-tail recognition is that the distribution shift is implicit. One of the main challenges in applying energy regularization is determining the coefficient $\beta$ in Eq. 5 as we do not know the testing data distribution. In this paper, we introduce the influence function to determine the coefficient $\beta$. Influence function (Cook & Weisberg, 1982) was used to determine the influence of training data samples on the model performance (Koh & Liang, 2017). Similar to that in Koh & Liang (2017), given a training set with $n$ data points $D_{train} = \{(\mathbf{x}_1, y_1), (\mathbf{x}_2, y_2), \cdots (\mathbf{x}_n, y_n)\}$, the optimal parameter for empirical risk is given by $\hat{\theta} \overset{def}{=} \arg\min_\theta \frac{1}{n} \sum_{i=1}^n \mathcal{L}(\mathbf{x}_i, y_i, \theta)$. When we add an energy regularization on a training data point $(\mathbf{x}, y)$ with a small $\epsilon$, the new optimal parameter becomes $\hat{\theta}_{\epsilon,(\mathbf{x},y)} = \arg\min_\theta (\frac{1}{n} \sum_{i=1}^n \mathcal{L}(\mathbf{x}_i, y_i, \theta) + \epsilon E_\theta(\mathbf{x}))$. Assume that the empirical risk is twice-differentiable and strictly convex w.r.t. $\theta$, the influence function provides the influence of the energy regularization on $(\mathbf{x}, y)$:

$$
\mathcal{I}_{\hat{\theta}}(\mathbf{x}, y) \overset{def}{=} \frac{\mathrm{d}\hat{\theta}_{\epsilon,(\mathbf{x},y)}}{\mathrm{d}\epsilon}\Big|_{\epsilon=0} = -H_{\hat{\theta}}^{-1} \nabla_\theta E_\theta(\mathbf{x}).
\tag{8}
$$

where $H_{\hat{\theta}} = \frac{1}{n} \sum_{i=1}^n \nabla_\theta^2 \mathcal{L}(\mathbf{x}_i, y_i, \hat{\theta})$ is the Hessian matrix. Using the chain rule, the influence on the loss at a validation point $(\mathbf{x}_{va}, y_{va})$ is:

$$
\begin{aligned}
\mathcal{I}_{(\mathbf{x}_{va}, y_{va})}(\mathbf{x}, y) &\overset{def}{=} \nabla_\theta \mathcal{L}(\mathbf{x}_{va}, y_{va}, \hat{\theta})^\top \frac{\mathrm{d}\hat{\theta}_{\epsilon,z}}{\mathrm{d}\epsilon}\Big|_{\epsilon=0} \\
&= -\nabla_\theta \mathcal{L}(z_{test}, \hat{\theta})^\top H_{\hat{\theta}}^{-1} \nabla_\theta E_\theta(\mathbf{x}).
\end{aligned}
\tag{9}
$$

This definition is similar to that in Koh & Liang (2017). Based on our devised influence function of energy, we propose a principled method that introduces **I**nfluence **A**ware **E**nergy **R**egularization (we refer to it as **IAER**). It firstly calculates the average influence of energy regularization on a validation set $D_{val} = \{(\mathbf{x}_i^{val}, y_i^{val})\}_{i=1}^m$ for each training data point. To fairly compare with previous works, we take subsets of the training set as the validation set without introducing any additional data. The average influence of energy regularization on the validation set is

$$
\mathcal{I}_{val}(\mathbf{x}_i) = \frac{1}{m} \sum_{j=1}^m \mathcal{I}_{(\mathbf{x}_j^{val}, y_j^{val})}(\mathbf{x}_i)
\tag{10}
$$

To reduce the loss on the validation set, we increase the energy on the data points with a positive influence of energy regularization and decrease the energy on those with a negative influence. Therefore, we finetune the model with energy penalties determined by the corresponding influence value. The coefficient $\beta_{\mathbf{x}_i}$ for data $\mathbf{x}_i$ is

$$
\beta_{\mathbf{x}_i} = -\gamma \cdot \mathcal{I}_{val}(\mathbf{x}_i) / \mathcal{I}_{val}^{max}.
\tag{11}
$$

Here $\mathcal{I}_{val}^{max}$ is the maximum absolute value of influence value over the train set $D_{train}$, and we set the hyperparameter $0 < \gamma < 1$, by which we make sure that the energy regularization does not interfere the optimization on cross-entropy.

Table 2: Average testing accuracy (%) of our method on Imbalanced CIFAR10/CIFAR100.

| Dataset | Imbalanced CIFAR10 | | | | Imbalanced CIFAR100 | | | |
|---|---|---|---|---|---|---|---|---|
| Imbalance type | long-tailed | | step | | long-tailed | | step | |
| Imbalance Ratio | 100 | 10 | 100 | 10 | 100 | 10 | 100 | 10 |
| ERM (Cao et al., 2019) | 70.36 | 86.61 | 63.30 | 84.27 | 38.32 | 56.59 | 38.55 | 54.70 |
| LDAM-DRW (Cao et al., 2019) | 77.16 | 87.62 | 75.36 | 87.42 | 42.04 | 56.67 | **45.36** | 56.84 |
| ERM + IAER | 75.83 | 87.02 | 71.33 | 85.62 | 39.60 | **57.59** | 39.12 | 55.10 |
| LDAM-DRW + IAER | **78.37** | **87.72** | **75.89** | **87.70** | **42.81** | 56.67 | 44.61 | **56.9** |

## 4.2 EXPERIMENTS WITH IAER IN DIFFERENT SCENARIO

### 4.2.1 LONG-TAIL RECOGNITION

We first conduct experiments on the long-tail recognition scenario to test our energy regularization method. We evaluate IAER on the imbalanced version of CIFAR10, CIFAR100 (Cui et al., 2019) and ImageNet-LT (Liu et al., 2019) that are artificially created with class imbalance and iNaturalist 2018 (Van Horn et al., 2018), a naturally long-tailed dataset. Experiments are conducted on imbalanced CIFAR following Cao et al. (2019) and on ImageNet-LT following Kang et al. (2020). For a fair comparison, the validation set used to calculate the influence function is sampled from the training set, and the models are not exposed to testing data during training.

**Results on CIFAR.** We evaluate IAER with ResNet-32 trained by: 1) Empirical risk minimization (ERM): with equal weight for each training data, and the model is trained to minimize the cross-entropy. 2) LDAM-DRW (Cao et al., 2019): LDAM introduces a label-distribution aware margin loss, enlarging the decision while DRW applies re-weighting or re-sampling after the last learning rate decay. Since ERM is the basic training algorithm and a typical baseline while LDAM-DRW achieves SOTA on imbalanced CIFAR datasets, we take these two methods as baselines.

CIFAR10 and CIFAR100 both contain $50,000$ images in training and $10,000$ images in testing with 10 and 100 classes, respectively. We construct the imbalanced version of CIFAR10 and CIFAR100 by reducing the number of images for each class. Two types of imbalance are considered: long-tailed imbalance (Cui et al., 2019) and step imbalance (Buda et al., 2018). For long-tailed imbalance, the number of data points follows an exponential decay across different classes. For step imbalance, data points in half of the classes are reduced to the same number while the number of data points in the other classes remains the same.

As shown in Table 2, our method could effectively boost the testing performance after only 5 epochs. The more imbalanced the IAER is, the more effective it is. Notably, IAER could greatly improve the ERM pre-trained model. For instance, IAER reduces the testing error of the ERM pre-trained model for $5.47\%$ (from $29.64\%$ to $24.17\%$) on long-tailed CIFAR10 with the imbalance ratio at 100. For CIFAR100, IAER also improves the testing performance for the ERM pre-trained model and improves the LDAM-DRW pre-trained model on long-tailed CIFAR100 with the imbalance ratio at 100 and step-imbalanced CIFAR100 with the imbalance ratio at 10. However, the improvement in imbalanced CIFAR100 brought by our IAER is much smaller than that of imbalanced CIFAR10. We conjecture that the calculated influence of energy regularization on our sampled validation set for CIFAR100 is less accurate since the number of images per class in CIFAR100 is much smaller than that of CIFAR10 *e.g.* only 5 images for the least frequent class when imbalance ratio is 100.

**Results on ImageNet-LT And iNaturalist 2018.** We evaluate IAER with ResNeXt-50 (Xie et al., 2017) pre-trained by the techniques and protocols proposed in Kang et al. (2020) where each model is divided into two parts: backbone and linear classifier. The protocol includes (1) Classifier Retraining (cRT): employ the backbone trained with ERM and retrain the linear classifier with the class balance sampling method. (2) Learnable Weight Scaling (LWS): Rescale the weight of the classifier for each class by a rescaling factor learned with the class balance sampling method as in cRT. ImageNet-LT (Liu et al., 2019) is artificially truncated from ImageNet (Deng et al., 2009), where the label distribution follows a long-tailed distribution. It has 1000 classes, and the number of images per class ranges from 1280 to 5. iNaturalist 2018 (Van Horn et al., 2018) is a real-world, long-tailed dataset with $8142$ classes. We follow Liu et al. (2019) and report the testing accuracy on three kinds of class sets: *Many-shot* (over 100 images), *Medium-shot* ($20 \sim 100$ images) and *Few-shot* (less than 20 images). The testing accuracy on all classes is denoted as *All*.

Table 3: Experiments on ImageNet-LT and iNaturalist. The validation set for IAER is composed of images in the train set.

| Dataset and model | ResNeXt-50 on ImageNet-LT | | | | ResNet-152 on iNaturalist | | | |
| Method | Many | Median | Few | All | Many | Median | Few | All |
|---|---|---|---|---|---|---|---|---|
| cRT (Kang et al., 2020) | 61.8 | 46.2 | 27.4 | 49.6 | 75.9 | 71.9 | 69.1 | 71.2 |
| cRT + IAER[Few] | 61.0 | 45.6 | **29.1** | 49.3 | 76.1 | 71.6 | **69.5** | 71.2 |
| cRT + IAER[Median] | 58.5 | **48.7** | 26.0 | 49.4 | 75.8 | **72.3** | 68.0 | 71.0 |
| cRT + IAER[Many] | **62.7** | 44.5 | 26.9 | 49.1 | **77.8** | 69.9 | 66.5 | 69.4 |
| LWS (Kang et al., 2020) | 60.2 | 47.2 | 30.3 | 49.9 | 74.3 | 72.4 | 71.2 | 72.1 |
| LWS + IAER[Few] | 60.1 | 47.1 | **32.1** | **50.1** | 74.4 | 72.4 | **71.6** | **72.3** |
| LWS + IAER[Median] | 58.1 | **49.0** | 30.3 | 50.0 | 74.5 | **72.8** | 70.9 | 72.2 |
| LWS + IAER[Many] | **61.5** | 45.5 | 30.1 | 49.6 | **74.9** | 72.6 | 71.0 | 72.2 |

Table 4: Experiment results for subpopulation shift conducted based on SubpopBench (Yang et al., 2023) on CMNIST (Arjovsky et al., 2019), MetaShift cats *vs.* dogs (Liang & Zou, 2022), NICO++ (Zhang et al., 2023), Waterbirds (Wah et al., 2011) and CivilComments (Borkan et al., 2019).

| Dataset and Method | ERM | | ERM + IAER | |
| Metric | Mean | Worst | Mean | Worst |
|---|---|---|---|---|
| CMNIST | 77.8 | 4.6 | **78.1** | **14.3** |
| MetaShift | 90.4 | 66.2 | **90.6** | **67.7** |
| NICO++ | 82.0 | 30.0 | **82.8** | **33.3** |

Note that the minimum number of data points per class in the training set of iNaturalist is 2, which makes the possible class-balanced subset of the training set extremely small. Therefore, for ImageNet-LT and iNaturalist, we take images of *Many-shot*, *Medium-shot*, and *Few-shot* classes in the training set as the validation set, respectively. We employ the backbone provided by Kang et al. (2020) and finetune or retrain the classifier. For ImageNet-LT, we calculate the influence of energy regularization on the ResNeXt-50 pre-trained for 90 epochs and retrain the classifiers with IAER for 10 epochs. For iNaturalist 2018, we calculate the influence of energy regularization of the ResNet-152 pretrained for 200 epochs. The classifiers on the iNaturalist are retrained for 30 epochs with energy regularization. More details are in Appendix A.

As shown in Table 3, IAER[Few] means that the validation set used for calculating influence is composed of images of *few-shot* classes in the training set while IAER[Median] and IAER[Many] means the validation set is composed by *median-shot* classes and *many-shot* classes respectively. For ImageNet-LT and iNaturalist 2018, we could observe that the accuracy on the classes used to calculate the influence function is boosted *e.g.* IAER[Few] boosts the accuracy of the classifier on *few-shot* classes while IAER[Median] boosts the accuracy of the classifier on *median-shot* classes. LWS combined with IAER[Few] could improve the accuracy of the whole testing set.

### 4.2.2 RESULTS FOR SUBPOPULATION SHIFT

We follow SubpopBench (Yang et al., 2023) to conduct experiments for subpopulation shift. Subpopulations are subgroups of a data domain divided based on certain features, and subpopulation shift is a type of distribution shift characterized by changes in the proportion of some subpopulations. The long-tail dataset could be seen as a special case of subpopulation shift. We conduct our experiments on widely used datasets in SubpopBench, including ColoredMNIST (Arjovsky et al., 2019), MetaShift cats *vs.* dogs (Liang & Zou, 2022), NICO++ (Zhang et al., 2023). For a fair comparison, we use the training set as the proxy validation set to calculate the influence function. For the model training, we follow the setting in SubpopBench. As shown in Table 4, both the mean accuracy and the worst accuracy are improved with our IAER.

### 4.2.3 RESULTS FOR DOMAIN GENERALIZATION

We follow DomainBed (Gulrajani & Lopez-Paz, 2021) to conduct experiments for domain generalization. DomainBed is a testbed for domain generalization. As Gulrajani & Lopez-Paz (2021) shows that ERM outperforms SOTAs by average performance on common benchmarks as evaluated

Table 5: Experiment results for domain generalization conducted on CMNIST (Arjovsky et al., 2019), PACS (Li et al., 2017) and VLCS (Fang et al., 2013).(* means we use the results from Domainbed (Gulrajani & Lopez-Paz, 2021))

| Method | CMNIST | PACS | VLCS |
|---|---|---|---|
| ERM (Vapnik, 1998) | $51.5 \pm 0.1$ | $85.5 \pm 0.1$ | $77.4 \pm 0.2$ |
| IRM (Arjovsky et al., 2019)* | $52.0 \pm 0.1$ | $83.5 \pm 0.8$ | $78.5 \pm 0.5$ |
| GroupDRO (Sagawa et al., 2019)* | $52.0 \pm 0.0$ | $84.4 \pm 0.8$ | $76.7 \pm 0.6$ |
| MLDG (Li et al., 2018a)* | $51.5 \pm 0.1$ | $84.9 \pm 1.0$ | $77.2 \pm 0.4$ |
| CORAL (Sun & Saenko, 2016) | $51.2 \pm 0.1$ | $86.1 \pm 0.2$ | $78.8 \pm 0.6$ |
| SagNet (Nam et al., 2021) | $51.7 \pm 0.0$ | $86.3 \pm 0.1$ | $77.8 \pm 0.5$ |
| ERM + IAER | $51.9 \pm 0.1$ | $86.6 \pm 0.1$ | $78.5 \pm 0.2$ |

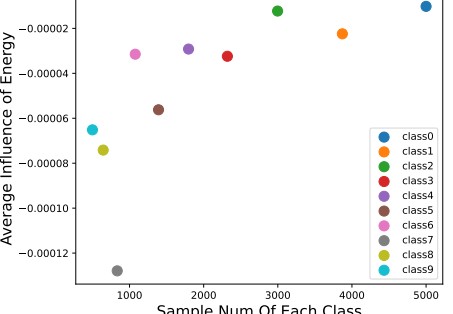
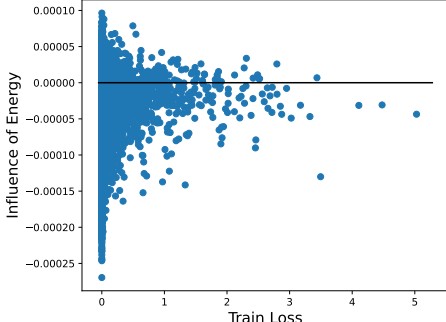

Figure 3: The average influence of energy regularization on CIFAR10-LT. Each dot represents a class, and the x-axis corresponds to the number of data samples.

Figure 4: Influence of energy regularization and loss of each training data point (a dot in the plot) in long-tailed CIFAR10 for ResNet-32 trained with ERM.

in a consistent and realistic setting, we combine our method with ERM and take the algorithms implemented in DomainBed as baselines. Experiments are performed on benchmarks ColoredMNIST (Arjovsky et al., 2019), PACS (Li et al., 2017) and VLCS (Fang et al., 2013).

As mentioned in Sec. 4.1, we take the training set as the validation set to calculate the influence function for a fair comparison. We follow the setting in DomainBed (Gulrajani & Lopez-Paz, 2021) to train the model. We demonstrate the results using training domain validation as model selection criteria, which use a validation set sampled from the training domain for model selection. For each algorithm and testing domain, we conduct a random search of 5 trails. For more experimental details, see Appendix A. As shown in Table 5, our IAER could improve the accuracy on the test domain without the test domain information. This implies that regularizing energy on the training domains helps generalize the training domains to the testing domain. More results are in Appendix B.

## 5 OTHER EMPIRICAL RESULTS

### 5.1 AVERAGE INFLUENCE OF ENERGY REGULARIZATION ON CIFAR10-LT

We plot the average influence of energy regularization of each class on CIFAR10-LT. As shown in Fig. 3, the average influence of energy regularization of different classes is positively correlated to the number of data points of the corresponding class (The Pearson's R is 0.70 for long-tailed CIFAR10 with imbalance ratio at 10). The lower the influence of the energy regularization, the lower the testing loss of the classifier would be after adding a positive energy regularization. It indicates that pushing down the energy value of data points of less frequent class and pulling up the energy value of data points of more frequent class would boost the testing performance *i.e.* pull up the predicted $\bar{p}(\mathbf{x})$ for less frequent class and push down the predicted $\bar{p}(\mathbf{x})$ for more frequent class. Since the probability density $p(\mathbf{x})$ of data points of less frequent class is much lower and the $p(\mathbf{x})$ of data points of more frequent class is much higher in the imbalanced training set compared to the

Table 6: Time required to approximate the influence function on different datasets.

| Model | Dataset | iteration | repeat times | time used |
|---|---|---|---|---|
| ResNet-32 | CIFAR10 | 5000 | 10 | 718.20s |
| ResNeXt-50 | ImageNet-LT | 1000 | 10 | 8492.92s |
| ResNet-152 | iNaturalist 2018 | 1000 | 10 | 9701.74s |

testing set, it shows that pushing the predicted $\bar{p}(\mathbf{x})$ closer to the real $p(\mathbf{x})$ of the testing set would boost the testing performance. According to Sec. 3.2, our regularization method generally enlarges the margin and down-weights for the less frequent classes. On the other hand, it reduces the margin and up-weights for the more frequent classes.

## 5.2 INFLUENCE OF ENERGY REGULARIZATION ON DATA SAMPLES WITH DIFFERENT LOSS

To investigate the relationship between the influence of energy regularization and the loss on each training data point, we provide a scatter figure as in Fig. 4. Note that the influence of energy regularization on the data points of similar training loss ranges from positive to negative. The influence of energy regularization is not related to the training loss (Pearson's R is $-0.04$). This indicates that one could not predict the influence of energy regularization on the data point based on training loss. However, the range of the influence of energy regularization expands as the training loss decreases. Therefore, data points where energy regularization has a large influence are generally well-classified data points with low loss. As pointed out in Remark 4.1, the energy value is unstable during the training, *we conjecture that the un-regularized energy value of well-classified data points is one of the possible reasons for the overfitting of the classifier*.

## 5.3 TIME COMPLEXITY ANALYSIS FOR THE APPROXIMATION OF INFLUENCE FUNCTION

The main overhead of the proposed IAER is calculating the influence function of energy regularization. We calculate the influence function with stochastic estimation (Cook & Weisberg, 1982) following Koh & Liang (2017). We implement the calculation of the influence function based on the Python package for calculating the influence function (Lo & Bae, 2022). For imbalanced CIFAR10 and imbalanced CIFAR100, the influence function is calculated with stochastic estimation for 5000 iteration and averaged over 10 trails. For ImageNet-LT and iNaturalist, we calculate the influence function only on the classifier, and the influence function is calculated with stochastic estimation for 1000 iteration and averaged over 10 trails.

Since we approximate the influence function using stochastic approximation, the time used to calculate the influence function is determined by the choice of hyperparameters. We tested the time cost for calculating the influence function for ResNet-32 on CIFAR10 with Intel(R) Xeon(R) CPU E5-2678 v3 @ 2.50GHz and one GeForce RTX 2080Ti for 5000 iteration and averaged ten times. Approximating the influence function for a 5000 iteration takes 718.20s on average. As shown in Table 6, we report the time used to approximate the influence function on different datasets for different models with the same hardware settings.

## 6 CONCLUSION

In this paper, we propose to regularize the energy among training data samples. We first show that various methods for long-tail recognition implicitly apply energy regularization, which pushes the energy distribution close to the test distribution. We further propose an influence-aware energy regularization for various OOD generalization scenarios, such as subpopulation shift and domain generalization. The main limitation comes from the use of the influence function. First of all, it requires a validation set. In this paper, we use a subset of the training set as the validation set. Experimental results show that the choice of the validation set largely affects the performance of IAER. Secondly, as defined for convex loss functions, the influence function may not reflect the actual influence for neural networks. The approximation of the influence function is also a bottleneck, which is hard to converge and requires high computational cost. We hope the idea of regularizing energy on training data samples will innovate future works.

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

## A    EXPERIMENT DETAILS

### A.1    DETAILS FOR THE CALCULATION OF INFLUENCE FUNCTION

The calculation of the influence function requires a validation set. For imbalanced CIFAR10 and CIFAR100, we sample data points of each class from the class-imbalanced training set to compose the validation set. The number of data points per class is determined by the minimum number of data points per class in the training set. For ImageNet-LT, since it has a validation split, we use the val split to calculate the influence function. For iNaturalist, since the minimum number of data points per class is APPall (2 images per class), we take the *few-shot* classes of the training set as the validation set.

We calculate the influence function with stochastic estimation (Cook & Weisberg, 1982) following Koh & Liang (2017). We implement the calculation of the influence function based on the Python package for calculating the influence function (Lo & Bae, 2022). For imbalanced CIFAR10 and imbalanced CIFAR100, the influence function is calculated with stochastic estimation for 5000 iteration and averaged over 10 trails. For ImageNet-LT and iNaturalist, we calculate the influence function only on the classifier, and the influence function is calculated with stochastic estimation for 1000 iteration and averaged over 10 trails.

### A.2    DETAILS FOR THE EXPERIMENTS ON IMBALANCED DATASET

We follow the setting in Cao et al. (2019) to train ResNet-32 on the imbalanced CIFAR dataset and report the performance of the model at the final epoch. The model is trained for 200 epochs with SGD optimizer where the learning rate is at $0.1$, momentum at $0.9$, and weight decay at $2e - 4$. The learning rate is decayed with factor $0.01$ at 160-th epoch and 180-th epoch. For IAER, we finetune the model for 5 epochs with batch size at $128$ and learning rate at $1e - 4$.

For ImageNet-LT and iNaturalist 2018, we employ the pre-trained model provided in Kang et al. (2020) and follow the setting in it to finetune the ResNeXt-50 (Xie et al., 2017) on ImgaeNet-LT and the ResNet-152 (He et al., 2016) on iNaturalist 2018. For ImageNet-LT, the classifier is finetuned for 10 epochs with batch size at $512$ and learning rate at $0.2$. For iNaturalist, the classifier is finetuned for 30 epochs with batch size at $512$ and learning rate at $0.2$.

The $\gamma$ in Eq. 9 is searched in $\{0.1, 0.5, 1, 10\}$ for CIFAR-LT and set to be $0.5$ for ImageNet-LT and iNaturalist2018. When the absolute value of energy regularization is bigger than the cross-entropy loss the $\gamma$ is set to be $\left\| \frac{\mathcal{L}_{ce}(\mathbf{x}_i^{tr}, y_i^{tr}, \theta)}{E_\theta(\mathbf{x}_i^{tr}) \cdot \mathcal{I}_{val}(\mathbf{x}_i^{tr}, y_i^{tr}) / \mathcal{I}_{val}^{max}} \right\|$

### A.3    DETAILS FOR THE EXPERIMENTS FOR DOMAIN GENERALIZATION

We follow the setting in Gulrajani & Lopez-Paz (2021) to conduct experiments and use the training domain validation set to calculate the influence function. The $\gamma$ in Eq. 9 is set to be $0.1$. We employ the same training settings and hyperparameters implemented in Gulrajani & Lopez-Paz (2021).

### A.4    COMPARISON BETWEEN THE INFLUENCE OF THE CROSS-ENTROPY AND THE INFLUENCE OF THE ENERGY.

The influence function of the cross-entropy has been widely used in data valuation (Koh & Liang, 2017) and active learning (Liu et al., 2021b). As shown in Fig. 5, the influence of the cross-entropy and the influence of the energy do not correlate with each other. The influence function of the cross-entropy as in previous works (Koh & Liang, 2017; Liu et al., 2021b) evaluates the influence of reweighting the data points. As for the influence function of the energy, we focus on the influence of the energy regularization, which acts as reweighting and margin control. Therefore there are fundamental differences between the influence of the cross-entropy and that of the energy.

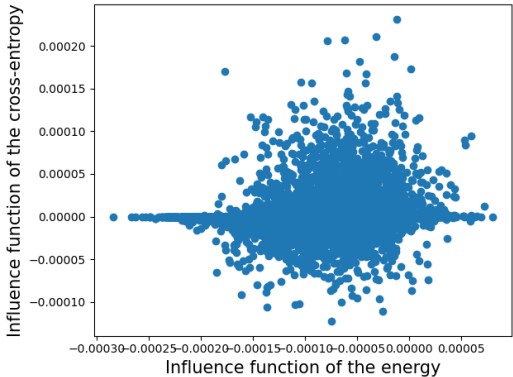

Figure 5: Relation between the influence of the cross-entropy and that of the energy. The influence is calculated for the ResNet-32 trained on CIFAR10 where each point represents a data sample.

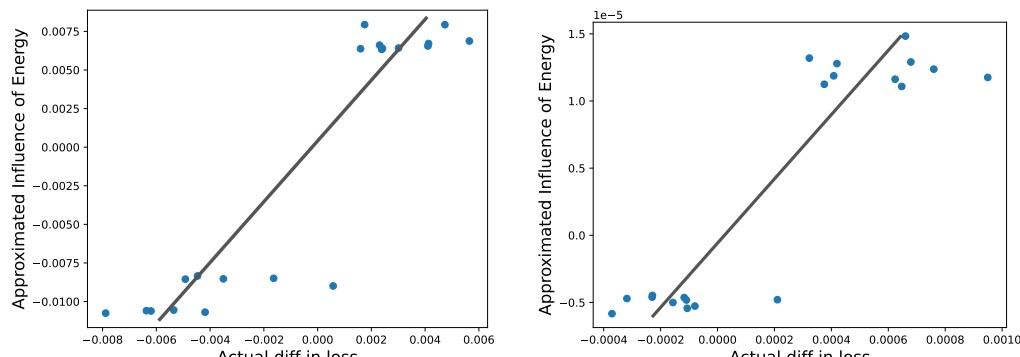

(a) ERM pre-trained model (Pearson's R = 0.9154)   (b) LDAM pre-trained model (Pearson's R = 0.9037)

Figure 6: The positive relation between the calculated influence function and the actual change in testing loss on pre-trained CNN. We plot the top 20 most influential data points.

# B   ADDITIONAL EXPERIMENT RESULTS

## B.1   ADDITIONAL RESULTS TO VALIDATE THE INFLUENCE FUNCTION OF ENERGY REGULARIZATION

In addition to Fig. 6, where we calculate the average influence over the whole testing set, we arbitrarily pick a wrongly classified data point and calculate its influence function following Koh & Liang (2017).

We calculate the influence function for the ResNet-32 trained with ERM on the long-tail CIFAR10 and long-tail CIFAR100, where the imbalance ratio is set to be 100. We plot the influence of energy regularization and the actual change in testing loss after finetuning the model with energy regularization for 50 epochs on 100 most influential data point. The calculated influence function has a positive relation to the actual change in loss (Pearson's R is 0.7875 on long-tail CIFAR10 and is 0.5744 on long-tail CIFAR100)

## B.2   ENERGY DISTRIBUTION SHIFTS DURING TRAINING

We plot the energy distribution of the training set during the training of ResNet-32 by ERM on the long-tailed CIFAR10 with an imbalance ratio of 100. As shown in Fig. 7, the energy distribution keeps changing during the training even though the training loss is stable *e.g.* from 100-th epoch to 150-th epoch. We further plot the distribution at 160, 170, 180, 190, and 200 epochs after the learning rate has decayed. As shown in Fig. 7(b), the energy distribution of the training set still changes when the learning rate is decayed and the model is converged.

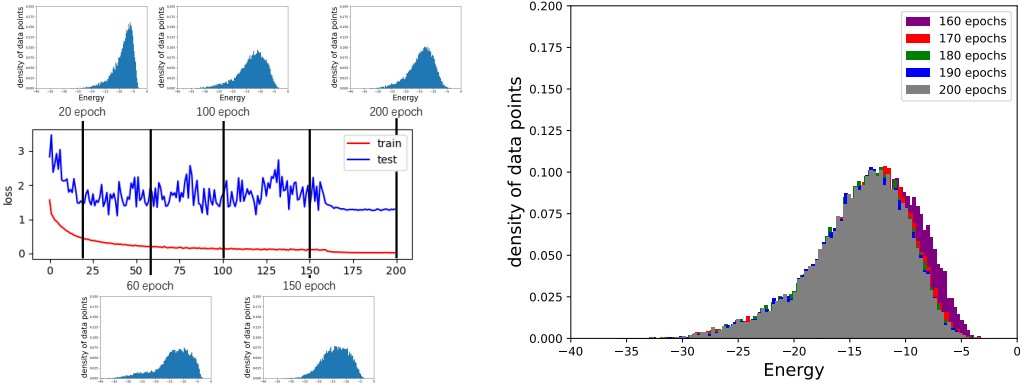

Figure 7: The energy distribution of a ResNet-32 trained on the long-tailed CIFAR10. (a): the energy distribution of the training set on different epochs during the training (b): the energy distribution of the training set at 160, 170, 180, 190, and 200 epoch.

| Model | Dataset | iteration | repeat times | time used |
|---|---|---|---|---|
| ResNet-32 | CIFAR10 | 5000 | 10 | 718.20s |
| ResNeXt-50 | ImageNet-LT | 1000 | 10 | 8492.92s |
| ResNet-152 | iNaturalist 2018 | 1000 | 10 | 9701.74s |

Table 7: Time required to approximate the influence function on different datasets for different models.

| Method | Testing accuracy on different test domains(%) | | | |
|---|---|---|---|---|
| Colored MNIST | +80% | +90% | -90% | |
| ERM | $72.3 \pm 0.0$ | $72.0 \pm 0.1$ | $10.1 \pm 0.1$ | |
| ERM +IAER | $73.8 \pm 0.1$ | $71.5 \pm 0.1$ | $10.6 \pm 0.1$ | |
| PACS | A | C | P | S |
| ERM | $85.6 \pm 0.1$ | $79.7 \pm 0.2$ | $98.9 \pm 0.1$ | $78.0 \pm 0.1$ |
| ERM + IAER | $84.2 \pm 0.1$ | $84.8 \pm 0.1$ | $97.3 \pm 0.1$ | $80.2 \pm 0.1$ |

Table 8: The detailed results for domain generalization where we report the testing accuracy(%) for different test domains.

### B.3 TIME COMPLEXITY ANALYSIS

The main overhead of the proposed IAER is calculating the influence function of energy regularization. Since we approximate the influence function using stochastic approximation, the time used to calculate the influence function is determined by the choice of hyperparameters. We tested the time cost for calculating the influence function for ResNet-32 on CIFAR10 with Intel(R) Xeon(R) CPU E5-2678 v3 @ 2.50GHz and one GeForce RTX 2080Ti for 5000 iteration and averaged ten times. Approximating the influence function for a 5000 iteration takes 718.20s on average. As shown in Table B.3, we report the time used to approximate the influence function on different datasets for different models with the same hardware settings.

### B.4 DETAILED RESULTS FOR DOMAIN GENERALIZATION

We report the detailed results of our experiments for domain generalization, as shown in Table 8.

### B.5 PROOF FOR REMARK 4.1

*Proof.* For the classifier $f_\theta : \mathbb{R}^D \to \mathbb{R}^K$, assume the energy on data point $\mathbf{x} \in \mathcal{R}^D$ is $E_\theta(\mathbf{x})$. For $\forall \mathcal{E} \in \mathcal{R}$, a classifier $g_\eta : \mathcal{R}^D \to \mathcal{R}^K$ could be defined that satisfies:

$$\forall i \in \{1, 2, \cdots, K\}, \quad g_\eta(\mathbf{x})[i] = f_\theta(\mathbf{x})[i]) - \mathcal{E} + E_\theta(\mathbf{x}). \tag{12}$$

By adding a certain value $E_\theta(\mathbf{x}) - \mathcal{E}$ to each logit $f_\theta(\mathbf{x})[i])$, the prediction of $g_\eta$ is the same as the prediction of $f_\theta$ while the energy value of $g_\eta$ is changed to $\mathcal{E}$.

For the predicted $\bar{p}_\eta(y|\mathbf{x})$ we have:

$$
\begin{aligned}
\bar{p}_\eta(y|\mathbf{x}) &= \frac{\exp\left[f_\theta(\mathbf{x})[y]\right) - \mathcal{E} + E_\theta(\mathbf{x})]}{\sum_i \exp\left[f_\theta(\mathbf{x})[i]\right) - \mathcal{E} + E_\theta(\mathbf{x})]}, \\
&= \frac{\exp\left[f_\theta(\mathbf{x})[y]\right)]}{\sum_i \exp\left[f_\theta(\mathbf{x})[i]\right)]}, \\
&= \bar{p}_\theta(y|\mathbf{x}).
\end{aligned}
\tag{13}
$$

For the energy $E_\eta(\mathbf{x})$ on the $g_\eta$, we have:

$$
\begin{aligned}
E_\eta(\mathbf{x}) &= -\log\sum_i \exp\left[g_\eta(\mathbf{x})[i]\right] \\
&= -\log\sum_i \exp\left[f_\theta(\mathbf{x})[i] - \mathcal{E} + E_\theta(\mathbf{x})\right] \\
&= -\log\left(\exp[E_\theta(\mathbf{x}) - \mathcal{E}] \cdot \sum_i \exp\left[f_\theta(\mathbf{x})[i]\right]\right) \\
&= \mathcal{E} - E_\theta(\mathbf{x}) - -\log\sum_i \exp\left[f_\theta(\mathbf{x})[i]\right] \\
&= \mathcal{E} - E_\theta(\mathbf{x}) + E_\theta(\mathbf{x}) \\
&= \mathcal{E}.
\end{aligned}
\tag{14}
$$

$\square$

