# OpenReview forum: "Regularizing Energy among Training Samples for Out-of-Distribution Generalization"
_ICLR.cc/2025/Conference — ICLR 2025 Poster_

### Official Review · Reviewer_aGfa · 2024-11-02

**Soundness:** 3
**Presentation:** 2
**Contribution:** 2
**Rating:** 6
**Confidence:** 2

**Summary:**

This paper proposes a new approach to improve model performance in detecting and generalizing to out-of-distribution (OOD) data. Unlike traditional methods, which primarily focus on the energy difference between IID and OOD data, this approach also considers energy differences within IID samples. The authors explain that the re-weighting and margin-based methods commonly used in long-tailed learning implicitly apply energy regularization. Then the authors introduce Influence-Aware Energy Regularization (IAER), which adjusts energy values based on influence functions derived from training data. Empirical experiments demonstrate the effectiveness of this method.

**Strengths:**

- The idea of considering energy differences among in-distribution data samples is both novel and sensible.
- The authors conduct a series of experiments, spanning long-tailed datasets, subpopulation shift benchmarks, and OOD generalization benchmarks, to evaluate the effectiveness of the proposed method.

**Weaknesses:**

- The connection between Sec.3 and Sec.4 appears somewhat disjointed. The motivation for the proposed method in Sec.4 seems detached from Sec.3, lacking continuity and a strong foundation.

- The explanation of the re-weighting and margin control methods as implicitly affecting training energy feels somewhat forced. For instance, the paper primarily uses the first term in Eq. (6) to justify the impact of $\hat{\beta}_x$. However, the second term in Eq. (6), which also contains $\hat{\beta}_x$, influences the final loss. Thus, the effectiveness of energy regularization remains somewhat ambiguous.

- The method's overall effectiveness is limited. For example, in Tab.3, when IAER is combined with cRT, the performance across all classes declines, and with LWS, the improvement brought by IAER across all classes is only about 0.2%.

- There are some typos, such as in Tab.3, where "iNatrualist" should be "iNaturalist".

**Questions:**

Please refer to the weakness section.

---

> ### Author Response · Authors · 2024-11-20
> **Rebuttal by authors**
>
> Dear reviewer aGfa,
>
> Thank you for your valuable feedback. We address each of your concerns and questions in the following.
>
> > The connection between Sec.3 and Sec.4 appears somewhat disjointed. The motivation for the proposed method in Sec.4 seems detached from Sec.3, lacking continuity and a strong foundation.
>
> In Sec.3, we theoretically show that energy reguarization can be decomposed into reweighting and margin control, two main stream of methods in long-tail recognition. We further empirically show that long-tail recognition methods apply implicit energy regularization.
> Based on the analysis in sec.3, in Sec.4, we propose a method to explicitly regularize the energy and extend the scenario from long-tail recognition to subpopulation shift and domain generalization.
>
> > The explanation of the re-weighting and margin control methods as implicitly affecting training energy feels somewhat forced. For instance, the paper primarily uses the first term in Eq. (6) to justify the impact of $\hat{\beta}_x$
> . However, the second term in Eq. (6), which also contains $\hat{\beta}_x$, influences the final loss. Thus, the effectiveness of energy regularization remains somewhat ambiguous.
>
> For Eq.6, the two terms both have coefficient $1-\hat{\beta}\_x$, which corresponds to data reweighting. The first term further have $\bar{p}(y|\mathbf{x}) -1$ changed to $\bar{p}(y|\mathbf{x}) -\frac{1}{1-\hat{\beta}\_{\mathbf{x}}}$ which corresponds to margin control, changing the margin defined as $f_θ (x)[y] − max_{j\neq y} f_θ (x)[j]$.
>
> To make Eq.6 more clear, we derive it as
> $\frac{\partial \mathcal{L}(\mathbf{x}, y, \theta)}{\partial \theta} = (1-\hat{\beta}\_{\mathbf{x}})\cdot \left( \left[\bar{p}(y|\mathbf{x}) -\frac{1}{1-\hat{\beta}\_{\mathbf{x}}} \right] \frac{\partial f\_{\theta}(\mathbf{x})[y]}{\partial \theta} + \sum\_{y'\neq y} \bar{p}(y'|\mathbf{x})\cdot \frac{\partial f\_{\theta}(\mathbf{x})[y']}{\partial \theta} \right).$
>
> where $\mathcal{L}(\mathbf{x}, y, \theta)=\mathcal{L}\_{ce}(\mathbf{x}, y, \theta) + \hat{\beta}\_{\mathbf{x}}\cdot E\_{\theta}(\mathbf{x})$
>
>
> > The method's overall effectiveness is limited. For example, in Tab.3, when IAER is combined with cRT, the performance across all classes declines, and with LWS, the improvement brought by IAER across all classes is only about 0.2%
>
> For iNaturalist and ImageNet-LT, the few-shot classes only have less than 20 images (the least class in iNaturallist only have 2 images), which makes it infeasible to sample a balanced validation set. Therefore, we approximate influence function on training data of *Many-shot*, *Medium-shot* and *Few-shot* classes emphasizing on part of the classes. In Tab. 3, we show that the energy regularization do affect the generalization ability.
>
> > There are some typos, such as in Tab.3, where "iNatrualist" should be "iNaturalist".
>
> Thank you for the feedback, we have modified it in our paper.
>
> We hope the response have addressed your concerns. We are looking forward to your reply.

---

> > ### Comment · Reviewer_aGfa · 2024-11-26
> >
> > I appreciate the authors' efforts in addressing the concerns raised in the rebuttal. My questions have been resolved. I decide to raise my score.

---

> > > ### Author Response · Authors · 2024-11-26
> > > **Thank you**
> > >
> > > Dear reviewer aGfa,
> > >
> > > Thank you for the response. We are glad to hear that our rebuttal has addressed your concerns. Thanks for your valuable feedback and we will keep polishing our paper.
> > >
> > > Best regards,
> > >
> > > Authors.

---

### Official Review · Reviewer_f4ew · 2024-11-04

**Soundness:** 2
**Presentation:** 2
**Contribution:** 2
**Rating:** 6
**Confidence:** 3

**Summary:**

The paper addresses the issue of OOD generalization in machine learning models. It proposes a novel approach to regularize the energy among ID data samples, which is crucial for improving OOD generalization performance. The authors argue that previous methods have focused on the energy difference between ID and OOD data, neglecting the energy differences among ID data samples. They introduce an energy regularization framework that unifies various long-tail recognition methods and extends it to OOD generalization scenarios. The paper empirically and theoretically demonstrates the effectiveness of the proposed method, called Influence-Aware Energy Regularization (IAER), through experiments on long-tail datasets, subpopulation shift benchmarks, and OOD generalization benchmarks

**Strengths:**

1. The paper offers a novel perspective on energy-based models by focusing on the energy differences among ID data samples, which is often overlooked in favor of the energy difference between ID and OOD data.
2. It provides a unified framework that connects long-tail recognition methods like data re-weighting and margin control, interpreting them as implicit energy regularization techniques.
3. The paper supports its claims with extensive experiments on various benchmarks, datasets, and scenarios, demonstrating the effectiveness of the proposed IAER method.

**Weaknesses:**

1. Inconsistent symbol usage can lead to ambiguity. For instance, in line 150, the variable $y$ denotes the specific label associated with a data point $x$, whereas in Equation 2, $y$ appears to represent the set of all training classes.
2. The definition of symbols should be clear and unambiguous. For example, in line 210, it states "for a data sample $x$ suppose the ground truth is $\hat{y}$".  The term"suppose" typically implies an assumption or prediction, which might be misleading in this context. Additionally, it is important to clarify the distinction between $y$ and  $\hat{y}$.
3. The baselines used in the study appear to be outdated. For instance, the baseline in Table 2 dates back to 2019, and the baseline in Table 3 is from 2020. Given the rapid advancements in the field, using more recent benchmarks would provide a more accurate and relevant comparison, thereby enhancing the validity and reliability of the results.

**Questions:**

1. In the transition from Eq. 5 to Eq. 6, the term $\hat{\beta_x} \cdot E_\theta(x)$ appears to be missing on the left side of the equation. Could the authors please clarify the derivation process between these two equations？Additionally, what does  $\mathcal{L}$ refer to in Eq.6?  Is $\mathcal{L}$ the same as $\mathcal{L}_{ce}$ in Eq.5?

2. Is the influence function calculated at each training epoch? Table 6 shows the time used for the influence function calculation, which appears to be computationally intensive. Could the authors provide a breakdown of the computational costs, specifically comparing the time spent on influence function calculation to the overall training time? This would give a clearer picture of the method's efficiency.

3. In Tab. 3, the selection of the validation set is crucial for model performance, especially in long-tailed scenarios where the test distribution can significantly differ from the training distribution. Could the authors provide a more detailed explanation of their validation set selection strategy, including the rationale behind their choices for different datasets? In Appendix A.1, why is a balanced validation set maintained for CIFAR, and how does the number of data points in the validation set affect the IAER? For ImageNet-LT, the validation set samples are excluded from the training set, whether it break the imbalance factor? Furthermore, excluding certain specific classes will downgrade the corresponding performance in Tab. 3, decreasing the generality of the method.

4. In line 317, the terms "positive influence" and "negative influence" are used. Could the authors provide clear definitions of these terms in the context of their method? Additionally, could they explain how these influences relate to the energy values of data points, particularly for low-energy points?

---

> ### Author Response · Authors · 2024-11-20
> **Rebuttal by authors**
>
> Dear reviewer f4ew,
>
> thank you for your time devoted to the reviewing process and providing us the valuable and detailed feedback. In the following, we address each of your comments.
>
> > Inconsistent symbol usage and the definition of symbols should be clear and unambiguous.
>
> We have modified our paper. The modified part is in red. To better clarify the symbols, we use $y$ for the ground truth label and $y'$ for all the training classes. In line 150, we use $y'$ to represent all the labels. In line 210 we use $y$ to represent ground truth. Thank you for the feedback, we will keep polishing our paper.
>
> > The baselines used in the study appear to be outdated. For instance, the baseline in Table 2 dates back to 2019, and the baseline in Table 3 is from 2020. Given the rapid advancements in the field, using more recent benchmarks would provide a more accurate and relevant comparison, thereby enhancing the validity and reliability of the results.
>
> Generally, the energy regularization is orthogonal to most previous methods (because models of the same risk/performance can have arbitrary energy value on each data sample as indicated in Remark 4.1). For subpopulation shift, we use subpopbench (2023) to conduct experiments. For domain generalization, we follow the widely used domainbed to conduct our experiments.
>
> We will include comparison with more recent work in our paper. In the following, we provide some additional experimental results on PACS combining our energy regularization with more recent methods. We are still preparing the additional experiments and will update experimental results promptly.
>
> >Inconsistency from Eq.5 to Eq.6.
>
> Because Eq.6 is too long for one line, we use $\mathcal{L}(\mathbf{x}, y, \theta)$ for the $\mathcal{L}_{ce}(\mathbf{x}, y, \theta) + \hat{\beta}_{\mathbf{x}}\cdot E_{\theta}(\mathbf{x})$. We have further clarified this in our paper. We sincerely apprreciate your attentive feedback.
>
> > Is the influence function calculated every epoch?
>
> No, the influence function is calculated once for one training. Therefore, with limited the number of approximation iteration, the computation overhead is acceptable.
>
> > In Tab. 3, the selection of the validation set is crucial for model performance, especially in long-tailed scenarios where the test distribution can significantly differ from the training distribution. Could the authors provide a more detailed explanation of their validation set selection strategy, including the rationale behind their choices for different datasets?
>
> Yes, the selection of the validation set is crucial. For long-tail recognition like CIFAR datasets, we sample a balanced validation set from the training set. It is under the motivation to make the model more balanced over classes, same as some previous works reweighting data points. However, for extremely long-tailed dataset like iNaturalist, the few-shot classes have less than 20 images, which  makes sampling a balanced validation set infeasible. Therefore, we take the training data of each *Many-shot*, *Medium-shot* and *Few-shot* classes to calculate influence and as shown in Tab.3, the choice of validation set do affect model performance.
>
> For domain generalization, because we have no access to the test domain data, we sample from the training domains to construct validation set.
>
> > In line 317, the terms "positive influence" and "negative influence" are used. Could the authors provide clear definitions of these terms in the context of their method? Additionally, could they explain how these influences relate to the energy values of data points, particularly for low-energy points?
>
> In our method, we use influence (approximated influence function) to determine the coefficient for energy regularization. Positive influence generally indicates the energy regularization would encourage a higher energy (lower probability) while negative influence indicates the energy regularization would ecourage a lower energy (higher probability).
>
> We hope the response have addressed your concerns. We are looking forward to your reply. Thank you for your valuable feedback.

---

> ### Comment · Reviewer_f4ew · 2024-11-27
>
> Thanks for the response. My concerns have been addressed. I decide to raise my score to 6.

---

> > ### Author Response · Authors · 2024-11-27
> > **Thank you**
> >
> > Dear Reviewer f4ew
> >
> > We are glad to hear that our rebuttal has resolved your concerns. We will keep on polishing our paper. Thanks for the valuable feedback.
> >
> > Best regards,
> >
> > Authors

---

### Official Review · Reviewer_16Hz · 2024-11-04

**Soundness:** 3
**Presentation:** 3
**Contribution:** 3
**Rating:** 6
**Confidence:** 4

**Summary:**

This submission presents a unified framework for energy-based models applied to both in-distribution and out-of-distribution (OOD) data. While prior methods in OOD detection focus on energy differences between in-distribution and OOD samples, this work highlights the importance of examining energy differences within in-distribution samples. The submission proposes investigating these differences, showing that techniques commonly used for subpopulation shifts—such as data re-weighting and margin control in long-tail classification—implicitly apply energy regularization. This framework is further extended to address more implicit distribution shifts, as seen in OOD generalization scenarios, using influence functions to support the approach. Experimental results on long-tail, subpopulation shift, and OOD generalization benchmarks demonstrate the effectiveness of this energy regularization method. The source code will be publicly available.

**Strengths:**

+ To my knowledge, this work is the early work that focuses on the energy of in-distribution data and use it as a regularization for learning models.
+ The Figure 2 well illustrates the motivation of this submision: the existing methods on long-tail problem can be reviewed as a kind of energy regularization to learn models.
+ The writing is clear. The related work, the derivation of unified view are also clear.
+ The expeiments domonstrate that proposed IAER can bring some improvements over baselines.

**Weaknesses:**

- The main concern lies with the experimental settings. Although the proposed IAER method shows some improvement over the standard baseline (ERM), it does not achieve competitive results compared to state-of-the-art methods. Specifically, Table 2 only includes comparisons with two works from 2019. Why are more recent works not included for comparison? This same question applies to Tables 3, 4, and 5. I would except more recent work to be included and disucssed. Just name a few: 1) Gradient Reweighting: Towards Imbalanced Class-Incremental Learning  2) Dynamic residual classifier for class incremental learning, 3) Towards principled disentanglement for domain generalization. 4) Flatness-aware minimization for domain generalization


- Lines 270–282 lack clarity. For instance, the statement “The least we can do is to prevent the model from being over-confident… note that the energy does not correspond to the prediction of the classifier (Remark 4.1), which is the reason most previous works overlook the energy disparity” requires further clarification: 1) yes, we usually do not have knowledge of test distribution. But how to prevent it from being over-confident? 2) I am not clear why Remark 4.1 explains why most previous works overlook the energy disparity between training data samples. 3) Additionally, the main point of Remark 4.1 is unclear. Could you clarify its purpose and significance?

**Questions:**

Please clarify the experimental analysis and Remark 4.1. Please see the above for more details.

---

> ### Author Response · Authors · 2024-11-20
> **Rebuttal by authors**
>
> Dear Reviewer 16Hz, thank you for your time and effort in providing us the valubale feedback. In the following We address each of your concerns.
>
> > The main concern lies with the experimental settings. Although the proposed IAER method shows some improvement over the standard baseline (ERM), it does not achieve competitive results compared to state-of-the-art methods. Specifically, Table 2 only includes comparisons with two works from 2019. Why are more recent works not included for comparison? This same question applies to Tables 3, 4, and 5. I would except more recent work to be included and disucssed. Just name a few: 1) Gradient Reweighting: Towards Imbalanced Class-Incremental Learning 2) Dynamic residual classifier for class incremental learning, 3) Towards principled disentanglement for domain generalization. 4) Flatness-aware minimization for domain generalization
>
> Thanks for the advise, we will include more recent works in our paper. For subpopulation shift, we use subpopbench (2023) to conduct experiments. For domain generalization, we follow the widely used domainbed to conduct our experiments, which is also compatible with many recent works such as the mentioned DDG method.
>
> As we are still preparing the additional experiments, we will update experimental results promptly.
>
> > Lines 270–282 lack clarity. For instance, the statement “The least we can do is to prevent the model from being over-confident… note that the energy does not correspond to the prediction of the classifier (Remark 4.1), which is the reason most previous works overlook the energy disparity” requires further clarification: 1) yes, we usually do not have knowledge of test distribution. But how to prevent it from being over-confident? 2) I am not clear why Remark 4.1 explains why most previous works overlook the energy disparity between training data samples. 3) Additionally, the main point of Remark 4.1 is unclear. Could you clarify its purpose and significance?
>
> Thanks for the good question, the following is our answer to the questions:
>
> The main point of Remark 4.1 is that models with identical classification performance (e.g. measured by cross entropy loss) could have different energy value on training data samples. It may also affect the generalization ability as shown in this paper.
>
> While we have witnessed many previous works focusing on regularizing the risk (classification performance) across different domains, the energy on training data samples has been overlooked.
>
> In preventing the model from being over-confident, we introduce energy regularization during training, which controls the energy (probability) assigned to different data samples.
>
> Once again, thank you for your valuable comments and support. We are more than happy to respond to any further questions.

---

> ### Comment · Reviewer_16Hz · 2024-11-27
>
> Thank you for your response. I have carefully reviewed the rebuttal as well as the comments from other reviewers. After consideration, I maintain my original score. As mentioned in my previous comments, this work is an early exploration of leveraging the energy of in-distribution data as a regularization for training models. I continue to encourage the authors to incorporate more recent works to further demonstrate the effectiveness of this regularization approach.

---

> > ### Author Response · Authors · 2024-11-27
> > **Thank you**
> >
> > Dear reviewer 16Hz,
> >
> > Thanks for your time devoted to the reviewing process and for providing us the valuable feedback and suggestions. We will continue to incorporate more recent works. Thanks again for your effort devoted to reviewing our paper.
> >
> > Best regards,
> >
> > Authors

---

### Official Review · Reviewer_GFVu · 2024-11-04

**Soundness:** 3
**Presentation:** 3
**Contribution:** 2
**Rating:** 6
**Confidence:** 3

**Summary:**

This paper proposes to explore the energy score between ID samples. It also theorectically suggests that previous methods for long-tail classification implicitly apply energy regularization. Some experiments are conducted to show the effectivness of the proposed engergy-based regularization.

**Strengths:**

1. The paper is well-written and well-structured.
2. Experiments are extensive to show the effectiveness of the method.
3. Theorectical analysis makes the paper more solid.

**Weaknesses:**

1. It seems that the proposed method does not perform consistently better than baseline methods and relatively unstable.

2. The designed influence function requires a large amount of time and computational recouces for approximating $\beta$.

3. In Fig 3, it can be observed that class 7 is a outlier. What cause this class significantly deviate the trend?

4. Some tables use testing error, while some use accuracy. It would be more readable to make them consistent.

**Questions:**

See weakness.

---

> ### Author Response · Authors · 2024-11-20
> **Rebuttal by authors**
>
> Dear Reviewer GFVu,
>
> we scincerely appreciate the time and effort you devote to the reviewing process. We address each point of your concerns in the comments below.
>
> > It seems that the proposed method does not perform consistently better than baseline methods and relatively unstable.
>
> In this paper, we use influence function to empirically bridge subpopulation shift methods with implicit energy regularization. The main limitation of the proposed method also comes from the use of influence function, where we **use traninig set as a proxy validation set for fair comparison and we limit the time consumption for influence function approximation.**
>
> Generally, for subpopulation shift tasks (long-tail recognition as a special case), our method is significantly better than baseline. For domain generalization, the proposed method also shows improvement. It is limited by the limitation of the influence function. However, since our method is orthogonal to existing algorithms (focusing on training energy regularization), we leave it for future works to further overcome the limitation of the influence function.
>
> We are preparing additional experiments and will update the experimental results promptly.
>
> > The designed influence function requires a large amount of time and computational recouces for approximating $\beta$
>
> Yes, approximating influence function is computational intensive. We provide a detailed time complexity analysis in Appendix B.3. Since the influence function only need to be approximated once, the time consumption is acceptable as we limit the approximation iteration.
>
> > In Fig 3, it can be observed that class 7 is a outlier. What cause this class significantly deviate the trend?
>
> The following is our conjecture.
> The class 7 in Fig. 3 corresponds to horses in CIFAR10. There are many other classes with similar background and similar object shape such as class 3 (cat), class 4 (deer), and class 5 (dog). With a very low influence function value, it indicates the model need to pay more attention to this class and increase the implicit probability of horses shown in the figure, which may explain why the class 7 is an outlier.
>
> > Some tables use testing error, while some use accuracy. It would be more readable to make them consistent.
>
> Sorry for the inconvenience. We have modified our paper and use testing accuracy in every table. The disparity is because in different scenarios we follow different papers to conduct experiments  such as long-tail classification, covariate shift and domain generalization.
>
> We hope the rebuttal can address your concerns and we are more than happy to answer further questions. We are looking forward to your reply.

---

> > ### Comment · Reviewer_GFVu · 2024-11-27
> >
> > Dear authors,
> >
> > Thank you for your efforts. I would like to maintain my positive rating.
> >
> > Best regards,
> >
> > Reviewer GFVu

---

> > > ### Author Response · Authors · 2024-11-27
> > > **Thank you**
> > >
> > > Dear Reviewer GFVu,
> > >
> > > Thank you for your time and your valuable feedback. We sincerely appreciate your effort devoted to the reviewing process. We will keep polishing our paper.
> > >
> > > Best regards,
> > >
> > > Authors

---

### Author Response · Authors · 2024-11-20
**Joint Response by Authors**

Dear AC and Reviewers,

We would like to express our scincere gratitude to the time and effort you dedicated to the reviewing process. We are delight to hear that reviewers find the paper to be novel (GFVu, 16Hz, f4ew, aGfa), well-written (GFVu, 16Hz) with extensive experiments (GFVu, 16Hz, f4ew, aGfa).

Generally, the main concern of reviewers focus on the outdated baselines. As we conduct experiments for subpopulation shift and domain generalization, we follow the widely used Subpopbench and Domainbed to conduct experiments. While DomainBed contains earlier methods, it is also compatible with more recent methods. We will add additional experiments regarding more recent methods. Currently, we are still preparing the experimental results, we will update the experimental results promptly.

Reviewers have provided many valuable and detailed comments. We have updated our paper according to these comments (in red) and we will keep polishing our paper.
To better address the comments and questions, we have posed a rebuttal for each reviewer. We look forward to your reply and are more than happy to respond to any further comments. Once again, thank you for your valuable comments and support.

Yours,

Authors.

---

> ### Author Response · Authors · 2024-11-23
> **Additional Experimental Results (1)**
>
> Dear Reviewers,
>
> We have conducted an additional experiment combining energy regularization with one of the recent domain generalization methods Risk Distribution Matching (RDM) [1]. We report the results on PACS with model selection using the training-domain validation set.
>
> | Method   | A    | C    | P    | S    | Avg  |
> | -------- | ---- | ---- | ---- | ---- | ---- |
> | RDM      | 86.6 | 82.2 | 96.2 | 80.7 | 86.4 |
> | RDM+IAER | 88.2 | 82.7 | 96.3 | 79.9 | 86.8 |
>
> While the time is limited, we are conducting more experiments and looking forward to your reply.
>
> Best regards,
>
> Authors
>
> [1] Nguyen, Toan, et al. "Domain Generalisation via Risk Distribution Matching." Proceedings of the IEEE/CVF Winter Conference on Applications of Computer Vision. 2024.

---

### Author Response · Authors · 2024-11-25
**Inquiry for post-rebuttal comments**

Dear Reviewers,

We appreciate your effort devoted to the reviewing process and the valuable feedback you provided. Since the discussion period is approaching its end, we are eager to hear from you whether our rebuttal has addressed your concerns.

Please check our modified manuscript, which has been uploaded to the system. https://openreview.net/pdf?id=Lbx9zdURxe

We are always available for further questions.

Best,

Authors

---

### Meta-Review · Area_Chair_pDDN · 2024-12-11

**Metareview:**

This paper proposes a novel method for regularizing energy among in-distribution (ID) data samples, which demonstrates significant effectiveness across multiple challenging tasks, including long-tail classification, subpopulation shift, and domain generalization. The work highlights the often-overlooked importance of energy differences within ID samples, presenting a compelling case for its role in improving generalization and robustness to out-of-distribution (OOD) scenarios. The method is theoretically well-founded, and all reviewers have provided positive ratings.

**Additional Comments On Reviewer Discussion:**

All reviewers have provided positive ratings.

---

### Decision · Program_Chairs · 2025-01-22

Accept (Poster)